# Ameliorative Effect of Dabigatran on CFA-Induced Rheumatoid Arthritis via Modulating Kallikrein-Kinin System in Rats

**DOI:** 10.3390/ijms231810297

**Published:** 2022-09-07

**Authors:** Mahmoud E. Youssef, Mustafa A. Abdel-Reheim, Mohamed A. Morsy, Mahmoud El-Daly, Gamal M. K. Atwa, Galal Yahya, Simona Cavalu, Sameh Saber, Ahmed Gaafar Ahmed Gaafar

**Affiliations:** 1Department of Pharmacology and Biochemistry, Faculty of Pharmacy, Delta University for Science and Technology, Gamasa 11152, Egypt; 2Department of Pharmacology and Toxicology, Faculty of Pharmacy, Beni-Suef University, Beni Suef 62521, Egypt; 3Department of Pharmaceutical Sciences, College of Clinical Pharmacy, King Faisal University, Al-Ahsa 31982, Saudi Arabia; 4Department of Pharmacology, Faculty of Medicine, Minia University, El-Minia 61511, Egypt; 5Department of Pharmacology and Toxicology, Faculty of Pharmacy, Minia University, El-Minia 61519, Egypt; 6Department of Biochemistry, Faculty of Pharmacy, Port Said University, Port Said 42515, Egypt; 7Department of Microbiology and Immunology, Faculty of Pharmacy, Zagazig University, Al Sharqia 44519, Egypt; 8Faculty of Medicine and Pharmacy, University of Oradea, P-ta 1 Decembrie 10, 410087 Oradea, Romania; 9Department of Pharmacology and Toxicology, Faculty of Pharmacy, Port Said University, Port Said 42511, Egypt

**Keywords:** dabigatran, rheumatoid arthritis, ACPA, Kallikrein-Kinin system, Bradykinin, RANKL

## Abstract

Rheumatoid arthritis is an autoimmune disease that affects joints, leading to swelling, inflammation, and dysfunction in the joints. Recently, research efforts have been focused on finding novel curative approaches for rheumatoid arthritis, as current therapies are associated with adverse effects. Here, we examined the effectiveness of dabigatran, the antithrombotic agent, in treating complete Freund’s adjuvant (CFA)-induced arthritis in rats. Subcutaneous injection of a single 0.3 mL dosage of CFA into the rat’s hind leg planter surface resulted in articular surface deformities, reduced cartilage thickness, loss of intercellular matrix, and inflammatory cell infiltration. There were also increased levels of the Anti-cyclic citrullinated peptide antibody (ACPA), oxidative stress, and tissue Receptor activator of nuclear factor–kappa B ligand (RANKL). Proteins of the kallikrein-kinin system (KKS) were also elevated. The inhibitory effects of dabigatran on thrombin led to a subsequent inhibition of KKS and reduced Toll-like receptor 4 (TLR4) expression. These effects also decreased RANKL levels and showed anti-inflammatory and antioxidant effects. Therefore, dabigatran could be a novel therapeutic strategy for arthritis.

## 1. Introduction

Chronic inflammation can trigger the immune system to attack healthy tissue and organs in the body [1,2]. When left untreated, prolonged chronic inflammation can increase the risk of diseases such as rheumatoid arthritis. The inflammatory state associated with rheumatoid arthritis is characterized by joint swelling, damage, and disability. Early diagnosis and therapeutic intervention are crucial to reducing the disease progression and development of irreversible disability [3]. Current therapies for rheumatoid arthritis include methotrexate (first-line drug), glucocorticoids, tumor necrosis factor α (TNFα) inhibitors, and Janus kinase inhibitors [4]. These drugs are commonly administered in combination to improve the patient’s condition and reduce the disease prognosis. However, chronic administration of these drugs could induce potentially serious adverse reactions due to their immunosuppressive effects, which could increase the risk of opportunistic infections such as tuberculosis [5]. Additionally, due to their slow effect, analgesic drugs are usually prescribed to reduce joint pain, which could lead to additional adverse effects [6]. Accordingly, the design of new therapies for arthritis is essential to avoid the serious side effects of current therapies and provide better disease control. 

Dabigatran is an anticoagulant drug that acts by direct inhibition of the thrombin active site, preventing the cleavage of fibrinogen into fibrin [7]. Dabigatran was prescribed to prevent stroke in arrhythmic patients [8], as well as to prevent pulmonary embolism and deep vein thrombosis [9]. Aside from the procoagulatory reducing effects in inflammatory conditions, dabigatran could also exhibit a marked suppressive effect on inflammatory cytokines and chemokine expression [10]. Moreover, dabigatran prevented the thrombin-induced expression of several inflammatory mediators and chemoattractant mediators, including monocyte chemoattractant protein-1 (MCP-1) [11], Interleukin 8 (IL-8), and chemokine (C-X-C motif) ligand 1 and 2 (CXCL1, and CXCL2) [12]. In low-density lipoprotein receptor-deficient mice, dabigatran reduced the polarization of M1 macrophages in visceral adipose tissue and the aortic wall by modulating the secretory profile of adipocytes [13]. The inflammatory-attenuating effects of dabigatran on arthritis have not been elucidated before.

Bradykinin is an activity plasma product of the KKS that is involved in various physiological and pathological conditions such as angiogenesis, coagulation, and inflammation [14,15,16]. The KSS system is composed of pre-kallikrein, factor XI, factor XII, and high-molecular-weight kininogen. Pre-kallikrein is activated to kallikrein, which triggers the cleavage of high-molecular-weight kininogen to form bradykinin [17]. Bradykinin is a potent inflammatory and vasodilator mediator [18]. It is believed that the activation of KKS participates in inflammatory responses and mediates the progression of arthritis [19]. Elevated levels of bradykinin and kallikrein and increased expression of the bradykinin receptor were detected in the synovial fluid of arthritic patients [20]. Furthermore, the promotive role of KKS and bradykinin in the progression of arthritis was reported in different models. The plasma levels of bradykinin were increased in rats insulted with streptococcal cell wall polymers, which led to polyarthritis [21]. The inhibition of KKS in rat models of arthritis minimized joint disability and reduced bradykinin plasma levels [22,23]. Additionally, the blocking of bradykinin receptors could inhibit arthritis [24]. Therefore, we examined the anti-inflammatory and KKS-suppressive properties of dabigatran in a CFA-induced arthritis rat model to characterize a novel therapeutic choice for arthritis.

## 2. Results

### 2.1. Histological Examination

Microscopic examination of normal knee joint samples showed normal histological structures of articular cartilage with an intact smooth surface, well-organized intact chondrocytes inside lacunae, either single or grouped at different zones (black arrow) with the obvious demarcation between calcified and non-calcified zones (arrowhead), and an intact homogenous intercellular matrix. Intact synovial membranes that covered the epithelium with minimal inflammatory cell infiltrates were observed (Figure 1a–c). Normal samples treated with dabigatran showed the same records as normal control samples without abnormal morphological alterations (Figure 1d–f). Examination of the joints of CFA-treated rats showed a wide erosion of the articular surface and irregularities with remarkable chondrocyte loss, as well as fibrous tissue replacement with significant inflammatory cell infiltrates (yellow arrow). Articular cartilage thickness significantly decreased with a remarkable loss of the intercellular matrix, as concluded by staining. Infiltration of mononuclear inflammatory cells in synovial membranes was clearly detected (Figure 1g–i). The arthritic group treated with a lower dose of dabigatran showed mild protective signs with evidence of minimized articular surface erosions and replacement with fibrous tissue (yellow arrow). However, evidence of higher chondrogenic activity was observed in deeper cartilaginous zones with apparent intact chondrocytes and more of the basophilic intercellular matrix in perilacunar zones (black arrow). Moderate persistence of subepithelial mononuclear inflammatory cell infiltrates was observed in synovial membranes (Figure 1j–l). The arthritic group treated with a higher dose of dabigatran showed persistence of minor articular surface fissures and irregularities (yellow arrow). However, significant thickness recovery of the articular cartilage and a well-defined calcified and non-calcified basophilic cartilaginous matrix (arrowhead) with abundant, more organized mature chondrocytes all over the articular surfaces were shown (black arrow). Moderate infiltrates of mononuclear inflammatory cells in synovial membranes were shown (Figure 1m–o).

### 2.2. Effect on ACPA and MDA

The injection of CFA increased ACPA and MDA levels (Figure 2a,b). Treatment with a low or high dose of dabigatran significantly suppressed both markers. High doses of dabigatran resulted in a significant drop in ACPA levels compared to the arthritic mice that received a lower dose of dabigatran. However, MDA levels did not show a significant difference between CFA+L/DABI and CFA+H/DABI groups (Figure 2a,b).

### 2.3. Effect on Kallikrein, Kallidin, and Bradykinin

Kallikrein levels were significantly elevated in the untreated arthritic group (Figure 3a,b,e). Treating arthritic rats with dabigatran significantly reduced kallikrein levels without reaching the levels of normal healthy rats. Similarly, kallidin and bradykinin levels were elevated after insult with CFA (Figure 3a,c–e). Daily treatment with different doses of dabigatran significantly decreased the levels of bradykinin and kallidin. A high dose of dabigatran reduced the tissue level of kallidin to restore it to the level of normal healthy rats. It is worth mentioning that there was no significant difference in kallidin and bradykinin levels between arthritic groups that were treated with low or high doses of dabigatran (Figure 3a,c–e), indicating that both mediators exhibit similar responses, irrespective of the dose of Dabigatran.

### 2.4. Effect on TLR4 and RANKL

The effect of daily administration of dabigatran on tissue levels of TLR4 and RANKL showed parallel results (Figure 4a,b). The untreated arthritic group showed an elevation in the tissue levels of both markers. Daily administration of either dose of dabigatran lowered TLR4 and RANKL levels without reaching normal levels. TLR4 and RANKL levels significantly decreased in arthritic rats that were treated with a high dose of dabigatran compared to the arthritic group that received lower doses of dabigatran (Figure 4a,b).

### 2.5. Effect on Collagen I

Single subcutaneous injection of CFA induced collagen formation in arthritic untreated rats (Figure 5a,b). Daily administration of dabigatran in CFA-treated rats significantly reduced the collagen expression to a level higher than the level of normal healthy untreated rats. The administration of higher doses of dabigatran significantly reduced collagen expression comparative to the arthritic group with lower doses (Figure 5a,b).

## 3. Discussion

Over the last few years, particular concern has been given to examining new therapies for rheumatoid arthritis as the current treatments are associated with severe adverse effects [25]. Therefore, our study focused on investigating the role of dabigatran, the anticoagulant drug, as a novel antiarthritic drug. 

CFA is experimentally used in most protocols for the induction of various autoimmune diseases including arthritis [26]. Subcutaneous injection of CFA results in hyperimmunization and increased circulating antigen-specific antibodies leading to joint disability and inflammation [27]. In rats, CFA induces arthritis in two phases, where periarticular inflammation is developed first and is followed by a phase of bone involvement that resembles what occurs in arthritis [26,28]. CFA was found to enhance the release of antibodies against collagen I and II, which are essential for bone regeneration [26,28,29]. The induction of arthritis leads to several pathological alterations including inflammatory synovitis with enhanced infiltration of inflammatory cells and an abrupt rise in the inflammatory mediators, followed by cartilage degradation and joint deformation [30,31]. The development of ACPA reflects the most specific autoimmunity marker for arthritis. The presence of ACPA in existing arthritis is linked to disease severity, whereas the production of ACPA during the early stages of arthritis could predict the disease prognosis. As a result, the formation of ACPA may be critical in the pathogenesis of arthritis [32]. MDA is an oxidative stress product that is associated with several diseases. The increase in MDA levels in the current model of arthritis could be linked to elevated ACPA levels [33] in addition to increased oxidative stress and lipid peroxidation in the synovial fluid of arthritic joints [34]. 

Pro-inflammatory kinins and cytokines seem to promote the development of rheumatoid arthritis. There is sufficient evidence showing that the kallikrein–kinin cascade has an important role in arthritis. Furthermore, there is a correlation between cytokines and kinins during inflammation. Kinins stimulate the production of cytokines, while cytokines have been found to enhance kinin activity. As a result, the inflammatory reaction may be increased and extended [35,36]. 

KKS is a cascade system divided into tissue KKS and plasma KKS [37]. In mammals, LMW kininogen is cleaved by kallikrein to form peptides, kallidin, and bradykinin. Both peptides activate B1 and B2 receptors [38]. Activation of the B2 receptor could enhance the formation of NO through the activation of the inducible nitric oxide synthase (iNOS) enzyme [39]. NO produced by chondrocytes is directly involved in arthritis, and its levels are commonly elevated in synovial fluid samples where it influences the inflammatory responses in joints and enhances cartilage destruction [40]. Moreover, the activation of B1 and B2 receptors increases TLR4 expression, which could lead to further activation of RANKL [41,42]. RANKL is a member of the TNF superfamily and is the regulator gene of apoptosis [43]. It also displayed a crucial role in bone growth and regulating bone remodeling at a physiological level [44]. RANKL is identified to regulate immune functions, where it was found in activated T-cells [45] and its role in arthritis was highlighted [46]. Therapeutic inhibition of RANKL is associated with reduced local bone erosion in animal models of arthritis [47]. 

Platelets are well known for their function in vascular homoeostasis; however, they also exhibit an important role in inflammation and immunomodulation [48]. There is growing evidence that platelets play a pathogenic role in autoimmune disorders and inflammatory arthritis, suggesting that coagulation and inflammatory systems are interconnected [49]. In rheumatoid arthritis, platelets that have been activated produce pro-inflammatory platelet microparticles that interact with leucocytes to cause systemic and joint inflammation. Immune complexes that activate platelets also stimulate dendritic cells to secrete interferon alpha (IFN-α), which is essential for the onset of systemic inflammation [50]. Herein, we indicate that the administration of dabigatran, the antithrombin anticoagulant agent, led to significant restoration of joint cartilage and enhanced the differentiation of mature chondrocytes all over the articular surface and reduced the infiltration of inflammatory cells. This improvement in histopathological features of joints was also reflected in the reported reduction in ACPA and MDA levels. Bone resorption could be stimulated by peptides of KKS and thrombin, the end-products of the coagulation cascade [51,52,53]. Kinins promote prostaglandin production in bone, which could lead to bone resorption [54]. Additionally, the resorptive effect of thrombin could be related to prostaglandin-dependent and independent pathways [55]. Furthermore, bradykinin and thrombin interact with IL-1 to promote bone resorption and prostaglandin production in a synergistic manner [54]. As a result, the coordinated action of thrombin and kinins may play a role in the rate of bone resorption in inflammation-induced bone loss. The inhibition of thrombin by dabigatran could decrease the activity of KKS leading to a further decrease in kinins production (Figure 6). This will decrease the activity of kallidin and bradykinin on B1 and B2 receptors. Decreased activity of B1 and B2 receptors will decrease the production of NO and reduce the expression of TLR4 [42,56]. The decrease in RANKL expression could be linked to the minimized activation of TLR4 as previously reported [57]. 

## 4. Materials and Methods

### 4.1. Animals

Adult female Wistar albino rats weighing 180 ± 20 g were purchased from the Modern Veterinary Office for Laboratory Animals, Cairo, Egypt. Housing, nutrition, acclimatization, handling, and experimental protocols of animals were monitored according to [58] after being approved under approval number ES23/2020 by the Ethics Committee of Faculty of Pharmacy, Minia University.

### 4.2. Chemicals, Antibodies, and Reagent Kits

CFA was ordered from Sigma-Aldrich, Mosby, MO, USA, and Dabigatran (Pradaxa)^®^ from Boehringer Ingelheim, Germany. The primary antibodies for immunoblotting, LMW kininogen (Ref. NBP1-96726), Bradykinin (Ref. NBP1-46328), Collagen (Ref. NB600-408), and beta-Actin AC-15 antibody (Ref. NB600-501), were purchased from Novus Biologicals, Centennial, CO, USA. The Kallikrein antibody (Ref. PA1709) was obtained from BOSTER BIOLOGICAL TECHNOLOGY, Pleasanton, CA, USA and species-specific secondary antibodies (horseradish peroxidase conjugated) were from GE Healthcare, Buckinghamshire, UK. Radio-Immuno-Precipitation Assay (RIPA) lysis buffer (Ref. 89900) was purchased from ThermoFisher Scientific, Rockford, IL, USA, and the RNA extraction and cDNA synthesis kits, as well as the RLT buffer, used for the quantitative polymerase chain reaction (qPCR) of RANKL and TLR4 were from Qiagen, Avenue Stanford, Valencia, CA, USA. Enzyme-linked immunosorbent assay (ELISA) kits for ACPA (Ref. MBS7240750) and malondialdehyde (MDA) (Ref. MBS268427) were purchased from MyBioSource, San Diego, CA, USA. All chemicals and reagents used were of analytical grade.

### 4.3. Experimental Approach

After one week of acclimatization, animals were divided into five groups: A control group receiving vehicles, an arthritis control group (receiving CFA only), a dabigatran group receiving vehicles plus a high dose of dabigatran, and two treatment groups receiving dabigatran (low and high doses) plus CFA. Test agents were administered daily for 14 days starting from day 12 after a single dose of CFA or the respective vehicle. Dabigatran was given at a dose of 10 and 30 mg/kg/day, p.o. [59]. 

### 4.4. Induction of Arthritis

Arthritis was induced by subcutaneous injection of a 0.3 mL single dose of CFA in the planter surface of the rat hind limb [58]. This model of arthritis was applied to female rats due to their high vulnerability to osteoarthritic pain and inflammation compared to male rats [60,61]. 

### 4.5. Serum Sampling

After the scheduled treatment period, a blood sample (2.5–3 mL) was collected on the last day of the experiment (14 days after CFS induction) by a non-heparinized capillary tube from the medial epicanthus of the animal’s eye in a sterile tube and left for 15 min followed by centrifugation for 30 min/4 °C at 1000× *g* in a cooling centrifuge (Model 3-30k, Sigma, Burlington, MA, USA). Serum was collected and stored at −20 °C for further biochemical assays.

### 4.6. Tissue Sampling

Rats were euthanized after being anesthetized with sodium thiopental (40 mg/kg), and then the whole knee joints, including synovium, adjacent tissues, and bones, were carefully exposed and separated. A single joint was kept at −80 °C for immunoblot analysis of tissue LMW kininogen, kallikrein, bradykinin, and collagen, as well as for qPCR analysis of tissue RANKL and TLR 4. The opposite joints were preserved for 72 h in formalin solution for histopathology. 

### 4.7. Analysis of Serum Biomarkers

The ELISA technique was employed to measure the serum levels of ACPA (sensitivity 0.06 ng/mL; Intra-assay Precision ≤ 8%) and MDA (sensitivity 0.05 nmol/mL; Intra-assay Precision ≤ 8%) according to the manufacturer’s instructions in a micro-titer plate reader ELISA Processing System (Model Stat Fax-2100, Awareness Technology, Palm City, FL, USA) based on Engvall [62]. 

### 4.8. Immunoblot Analysis of Tissue Biomarkers

Immunoblotting was performed as described earlier [63] for the analysis of tissue biomarkers (LMW kininogen, Kallikrein, Bradykinin and Collagen). Briefly, a part of the joint was lysed in RIPA buffer, kept in ice for 30 min, then centrifuged at 16,000× *g* in a cooling centrifuge to remove cell debris. Supernatants were transferred to new tubes and prepared for loading. Protein separation was conducted in a BioRad mini protein electrophoresis separation unit (Model 1658004, Sinorica International Patent and Trademark, Germantown, MD, USA), proteins were transferred to nitrocellulose membranes at 150 mA for 60 min, blocked with 5% non-fat milk in Tris-buffered saline (TBS), and underwent shaking for 60 min according to [64]. The membranes were incubated with primary antibodies against KLK1 (for Kallikrein), KNG1 (for LMW kininogen), BDKRB2 (for Bradykinin), COL1A1 (for Collagen), and β-actin (beta-Actin as a loading control) at dilutions of 1:5000 or 1:10,000 at 4 °C for at least 2 h, then washed with TBS-0.1%Tween 3 times (5 min each) and probed with the appropriate species-specific HRP conjugated secondary antibody at a dilution of 1:10,000 for 75 min. The signal was developed with the Tanon 5200 system after signal enhancement using an Enhanced Chemiluminescent detection (ECL) kit (Catalog #EK1002). 

qPCR analysis for the estimation of RANKL and TLR 4 expression in tissues was conducted with the aid of the Applied Biosystem Step One Plus Thermal Cycling Block (Model 4376600, ThermoFisher Scientific, Rockford, IL, USA). Briefly, total RNA was extracted from 30 mg of joint tissue using the RLT buffer according to the manufacturer’s protocol. The eluted RNA was used for cDNA synthesis and qPCR. Cycling conditions were as follows: Single denaturation cycle at 94 °C for 4 min, followed by 40 cycles of denaturation at 94 °C for 40 sec, annealing at 60 °C for 20 sec, and elongation at 72 °C for 30 sec. Primer sequences were as follows: RANKL, F 5′-CGCTCTGTTCCTGTACTTTCGAGCG-3′ and R 5′-TCGTGCTCCCTCCTTTCATCAGGTT-3′; TLR4, F 5′-CGCTTTCACCTCTGCCTTCACTACAG-3′ and R 5′-ACACTACCACAATAACCTTCCGGCTC-3′; β-actin, F 5′-GATTACTGCTCTGGCTCCTAGC-3′ and R 5′-GACTCATCGTACTCCTGCTTGC-3′. Relative mRNA expression levels were normalized to the housekeeping β-actin mRNA in the same sample using the 2^−ΔΔCq^ method according to [65,66].

### 4.9. Histopathology

After the decalcification of knee joints with EDTA, they were embedded in paraffin, then longitudinally sectioned at 5 μm and stained with hematoxylin and eosin (H & E) according to Youssef et al. [67,68].

### 4.10. Statistical Analysis

Data are represented as means ± standard errors of means (SEM) using GraphPad prism 9.0 software (GraphPad Software, San Diego, CA, USA). For statistical analysis, a one-way analysis of variance (ANOVA) test was performed followed by Tukey’s multiple comparisons test with the aid of the Statistical Package for Social Sciences (SPSS) software version 24 (San Diego, CA, USA).

## 5. Conclusions

The increased incidence of rheumatoid arthritis worldwide and the absence of efficient treatments with lower adverse effects have been of particular concern to find a new, more powerful medication with less adverse effects. In our study, dabigatran significantly enhanced the histopathological features of arthritic joints and improved their architecture. The inhibitory effect of dabigatran on thrombin led to the subsequent inhibition of KKS and reduced TLR4 expression. These effects also decreased RANKL and NO levels and showed anti-inflammatory and antioxidant effects. However, further investigations are needed to elaborate on other possible anti-arthritic mechanisms of dabigatran.

## Figures and Tables

**Figure 1 ijms-23-10297-f001:**
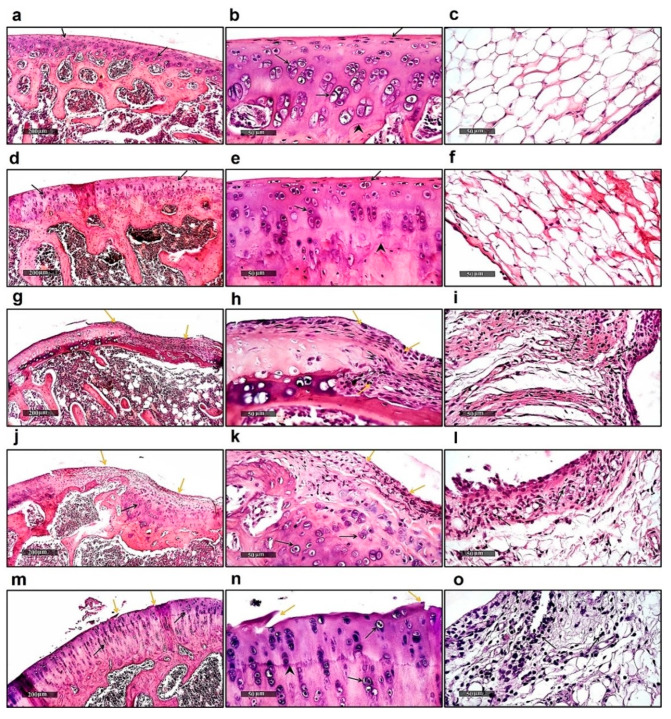
Histopathology. Photomicrographs of (**a**–**c**) knee joint samples of normal healthy rats showing well-organized intact chondrocytes inside lacuna (black arrow) in the presence of obvious demarcation between non-calcified and calcified zones (arrowhead); (**d**–**f**) knee joint sample of normal rats treated with dabigatran showing no anomalies; (**g**–**i**) knee joint samples of CFA-treated rats showing significant inflammatory cell infiltrates (yellow arrow) with a significant decrease in articular cartilage thickness; (**j**–**l**) knee joint samples of arthritic group treated with low dose of dabigatran showing minimized articular surface erosions and replacement with fibrous tissue (yellow arrow) with higher chondrogenic activity (black arrow); (**m**–**o**) knee joint samples of arthritic group treated with higher dose of dabigatran showing minor articular surface irregularities (yellow arrow), significant thickness recovery of articular cartilage (arrowhead), and the presence of mature chondrocytes all over articular surfaces (black arrow).

**Figure 2 ijms-23-10297-f002:**
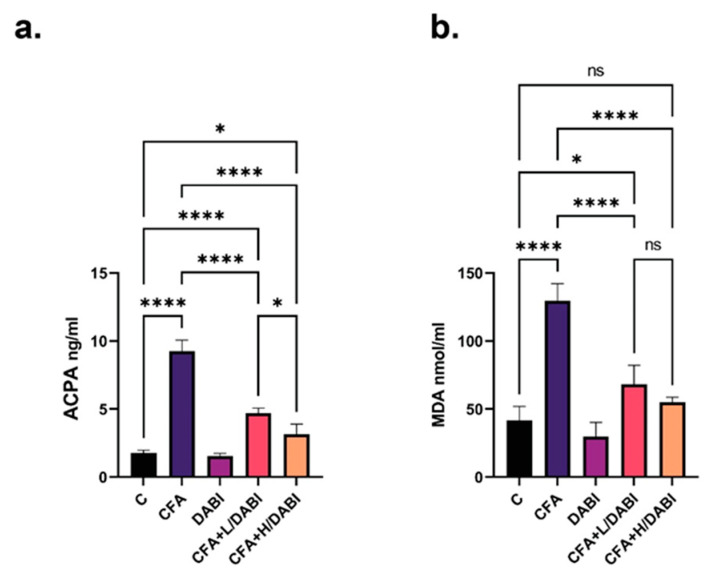
Effects of Dabigatran on serum ACPA and MDA levels. ELISA of serum ACPA levels (**a**), and MDA levels (**b**) in different groups as described in methods section. Not significant (ns); * *p* ≤ 0.05; **** *p* ≤ 0.0001, (*n* = 8).

**Figure 3 ijms-23-10297-f003:**
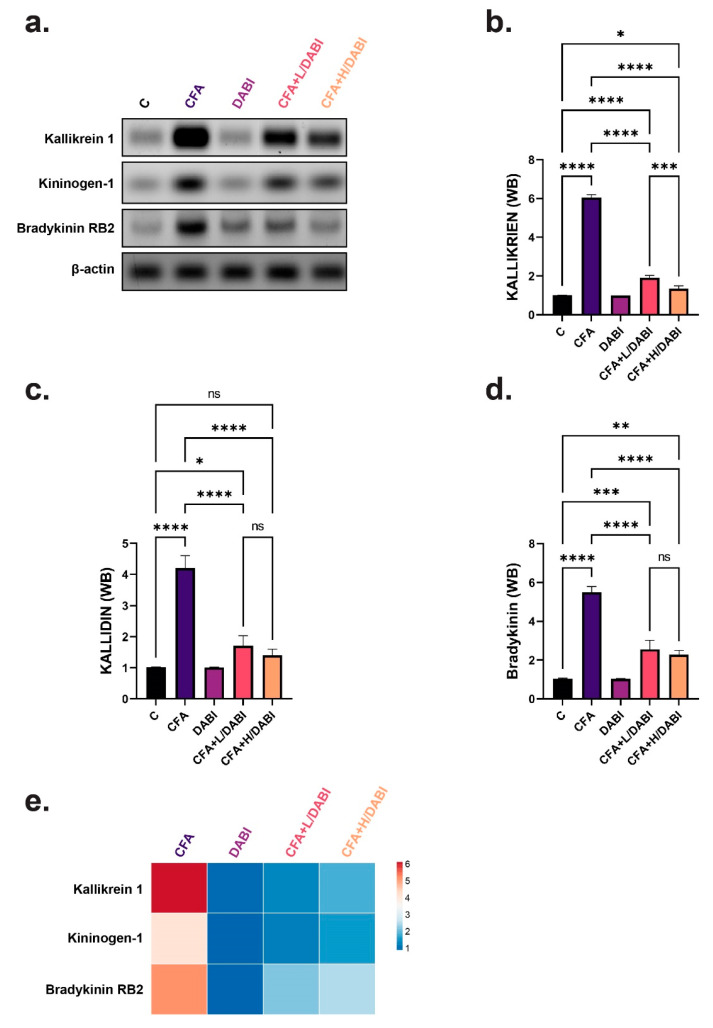
Immunoblot analysis of tissue biomarkers. (**a**) Immunoblots of Kallikrein-1, LMW kininogen-1, Bradykinin RB2, and beta-Actin as a loading control. (**b**) Chemiluminescent analysis of band intensities of Kallikrein-1 in (**a**). (**c**) Chemiluminescent analysis of band intensities of LMW kininogen-1 in (**a**). (**d**) Chemiluminescent analysis of band intensities of Bradykinin RB2 in (**a**). (**e**) Heatmap depicting trend of changes in tissue biomarkers analyzed by immunoblots in (**a**). Not significant (ns); * *p* ≤ 0.05; ** *p* ≤ 0.01; *** *p* ≤ 0.001; **** *p* ≤ 0.0001, (*n* = 8).

**Figure 4 ijms-23-10297-f004:**
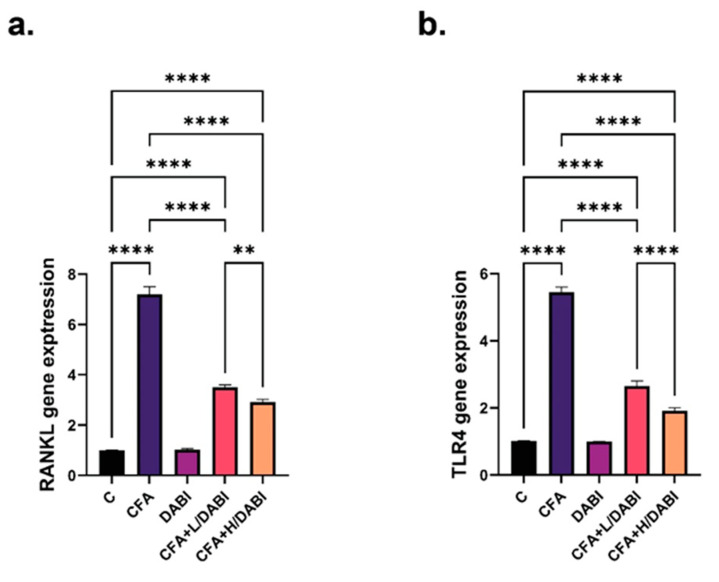
Effects of Dabigatran on RANKL and TLR4 expression. qPCR analysis of tissue RANKL (**a**), and TLR4 levels (**b**) in different groups as described in methods section. ** *p* ≤ 0.01; **** *p* ≤ 0.0001, (*n* = 8).

**Figure 5 ijms-23-10297-f005:**
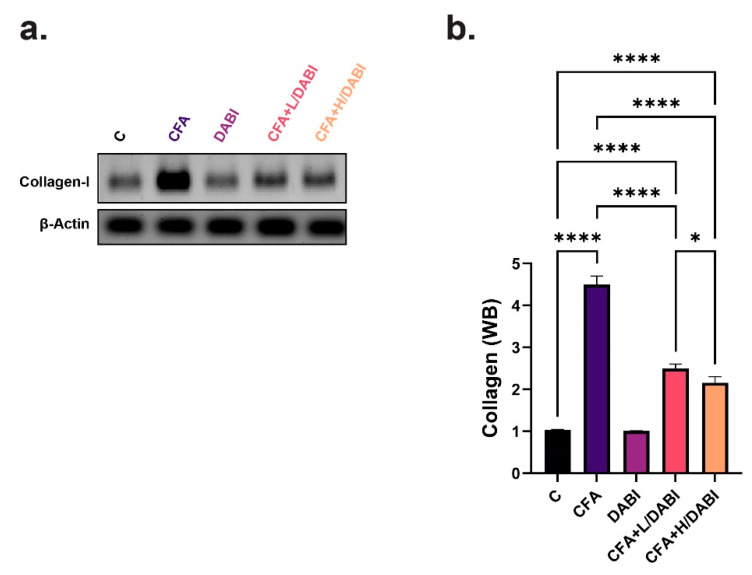
Immunoblot analysis of tissue Collagen. (**a**) Immunoblots of Collagen-1 and beta-Actin as a loading control. (**b**) Chemiluminescent analysis of band intensities of Collagen-1 in (**a**). * *p* ≤ 0.05; **** *p* ≤ 0.0001, (*n* = 8).

**Figure 6 ijms-23-10297-f006:**
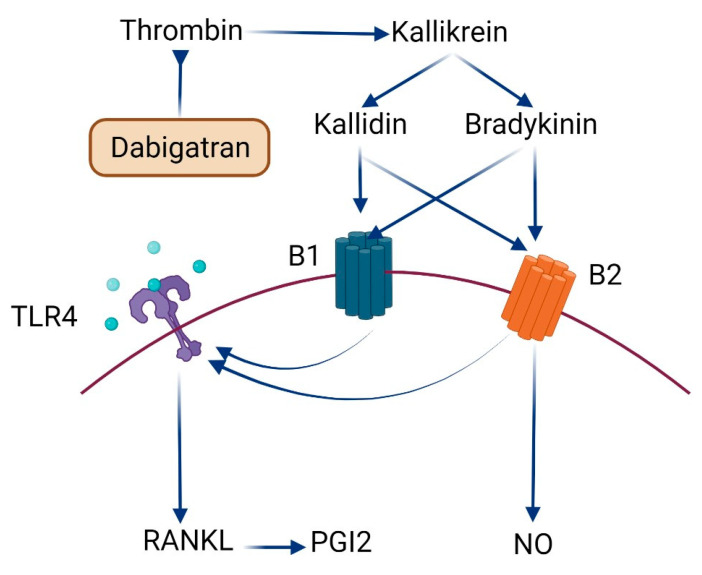
Schematic illustration summarizing the anti-inflammatory/antiarthritic roles of Dabigatran.

## Data Availability

Not applicable.

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
