# Peer review of "Ameliorative Effect of Dabigatran on CFA-Induced Rheumatoid Arthritis via Modulating Kallikrein-Kinin System in Rats"

_ijms, 2022, doi:10.3390/ijms231810297_

Round 1

Reviewer 1 Report

 The manuscript (ID: ijms-1890621) entitled “Ameliorative effect of dabigatran on CFA-induced rheumatoid arthritis via modulating kallikrein-kinin system in rats” submitted by the authors: Mahmoud E. Youssef, Mustafa A. Abdel-Reheim, Mohamed A. Morsy,Mahmoud El-Daly, Gamal M. K. Ata, Galal Yahya, Sameh Saber and Ahmed Gaafar submitted to section: Molecular Pathology, Diagnostics, and Therapeutics of the International Journal of Molecular Sciences could become a valuable contribution for the general readership of this journal after a major revision. Two important points have to be significantly improved.

The statistic part in the Material & Methods section is too thin and shows how cursorily this manuscript was created. The authors write that the only statistical method they used was ANOVA. However, this is by no means enough, because you then have to carry out further statistical analyzes based on the results of the ANOVA in order to check the data sets to be compared for statistical significance.

The refererence section is not complete. The authors should have a look at this paper and the literature cited there:

Lopatko Fagerström I, Ståhl AL, Mossberg M, Tati R, Kristoffersson AC, Kahn R, Bascands JL, Klein J, Schanstra JP, Segelmark M, Karpman D. Blockade of the kallikrein-kinin system reduces endothelial complement activation in vascular inflammation. EBioMedicine. 2019 Sep;47:319-328. doi: 10.1016/j.ebiom.2019.08.020. Epub 2019 Aug 20. PMID: 31444145; PMCID: PMC6796560.

Please consider also these two points:

(1) Why the fmale Wistar albino rats were chosen in the article and not male?

(2) The manuscript mentioned in the article that Dabigatran has anti-inflammatory and antioxidant effects, such as dabigatran can reduce the expression of RANKL and TLR4 and the level of ACPA and MDA in serum, but other anti-inflammatory and antioxidant indicators still need to be supplemented, and verified.

Author Response

Comment

The statistic part in the Material & Methods section is too thin and shows how cursorily this manuscript was created. The authors write that the only statistical method they used was ANOVA. However, this is by no means enough, because you then have to carry out further statistical analyzes based on the results of the ANOVA in order to check the data sets to be compared for statistical significance.

Response

Thank you for your comment. In statistical analysis we have performed Tukey’s posthoc test for multiple comparison and to evaluate the statistically significant difference between different groups.

Comment

The reference section is not complete. The authors should have a look at this paper and the literature cited there:

Response

Thank for your recommendation for this reference, the reference was cited in Introduction Section. Please, check the references list.

Comment

Why were the female Wistar albino rats chosen in the article and not male?

Response

It was previously reported that female rats are more vulnerable to osteoarthritic pain compared to male rats. We added this in Materials and Methods section

Please check the following references

Age and Sex Differences in Acute and Osteoarthritis-Like Pain Responses in Rats (nih.gov)

Female preponderance for development of arthritis in rats is influenced by both sex chromosomes and sex steroids - PubMed (nih.gov)

Comment

The manuscript mentioned in the article that Dabigatran has anti-inflammatory and antioxidant effects, such as dabigatran can reduce the expression of RANKL and TLR4 and the level of ACPA and MDA in serum, but other anti-inflammatory and antioxidant indicators still need to be supplemented and verified.

Response

Thank you for your comment. We highlighted different anti-inflammatory mechanism of dabigatran in the Introduction section, 2nd paragraph. In the Discussion section we focused on the KKS pathway and the modulating effect of dabigatran on this pathway.

Reviewer 2 Report

See annexed review report. 

Author Response

Comment

Abstract, line 33: The authors conclude that dabigatran mediates its effect through decreased RANKL expression and nitric oxide levels. However, the authors provide no data on NO production (nor expression of nitric oxide synthases).

Response

Thank you for your valuable comment. We did not evaluate the expression of NO, it was unintentionally added, so we deleted this information form the abstract

Comment

Materials & Methods or figure legends must include the days of blood sampling and euthanasia after CFA injection. Overall, the Materials & Methods should provide more details (qPCR primers and conditions, the sensitivity of the ELISA kits, etc.).

Response

The methods section is amended to include the missed data. Thank you for your comment

Comment

All figure legends must include the number of rats per group (n=?).

Response

We added the number of rats in all figure legends

Comment

Section 4.3, line 256: The authors state that they divided the animals into eight groups. Only five groups are described and studied in this study.

Response

We conducted our study on 5 different groups, we apologize for this mistake, and it was corrected in the manuscript.

Comment

Figure 6 should be revised. There are no data on B1 receptor expression, prostaglandin, and NO levels.

Response

Figure 6 explains the possible anti-inflammatory effects of dabigatran that is mediated by the inhibitory effect of dabigatran on KKS pathway. Some data were obtained from literature and all references were cited in the discussion section. Therefore, this figure could be considered as a graphical abstract or schematic illustration that summarizes the core of the discussion section.

Comment

Extending the discussion to the role of platelet activation in inflammatory arthritis would strengthen the manuscript. Furthermore, the authors must discuss the limitation of their study as thrombin action is not limited to activation of the kallikrein-kinin system. Protease-activated receptors and complement activation cascade also contribute to inflammation.

Response

Thank you for this valuable addition, we added a whole paragraph that highlights the role of platelet activation in inflammatory arthritis. In addition to the role of thrombin in the development of arthritis. Please check the discussion section: 5th and 6th paragraphs.

Comment

Line 201: …directly involved in in arthritis…

Response

This mistake was corrected

Author Response

Comment

This article investigated the effectiveness of dabigatran in treating complete Freund's adjuvant (CFA)-induced arthritis in rats and demonstrated that dabigatran could be a novel therapeutic strategy for arthritis.

Response

Thanks for your comment

Comment

This topic is original in the field, and it fills the gap of The inflammatory attenuating effects of dabigatran on arthritis.

Response

Thanks for your comment

Comment

There are some minor errors, for example, there are two “figure 2”.

Response

Thanks for your comment: we corrected this mistake as the following:

Figure 1. Histopathology.

Figure 2. Effects of Dabigatran on serum ACPA and MDA levels.

Comment

The conclusions are consistent with the current evidences.

Response

Thanks for your comment

Comment

The references are roughly appropriate. However, I am not sure if it is appropriate to refer to certain articles for general methods.

Response

Thanks for your comment. We have revised the references list, all cited references in the manuscript were consistent and appropriate with the given data. We updated the reference list to improve the quality of the manuscript.

Comment

Since you have demonstrated the effect of dabigatran on collage I, have you also tested the effect of dabigatran on collagen II and collagen X?

Response

The investigation of dabigatran effect on collagen II and collagen X would give a broader insight on dabigatran effects in arthritis. We believe that it is a limitation on our work and because of the promising results that we obtained in the current study, we are planning to conduct further studies on dabigatran anti-inflammatory studies in different inflammatory models.

Round 2

Reviewer 1 Report

The manuscript can now be published.

Reviewer 2 Report

I am satisfied with the author's responses to my questions/issues raised in my initial review.